# Lung Function Can Predict the Expected Inspiratory Airflow Rate through Dry Powder Inhalers in Asthmatic Adolescents

**DOI:** 10.3390/children9030377

**Published:** 2022-03-08

**Authors:** Roberto Walter Dal Negro, Paola Turco, Massimiliano Povero

**Affiliations:** 1Research & Clinical Governance, 37100 Verona, Italy; robertodalnegro@gmail.com (R.W.D.N.); turcop@libero.it (P.T.); 2AdRes Health Economics and Outcome Research, 10121 Torino, Italy

**Keywords:** DPIs, inspiratory airflow, intrinsic resistance, lung function, bronchial asthma, adolescents

## Abstract

Several factors affect drug delivery from dry powder inhalers (DPIs). Some are related to patient’s physiological characteristics, while others depend on DPIs’ technical aspects. The patient’s inspiratory airflow rate (IAR) affects the pressure drop and the turbulence needed to disaggregate the powder inside a DPI. The present study investigated whether lung function limitations occurring in asthmatic adolescents affect their IAR when inhaling through a DPI simulator. Eighteen consecutive adolescents with asthma were recruited, and IAR was randomly assessed at low-, mid-, and high-resistance regimens. A multiple logistic model was developed to evaluate the association of patients’ lung function characteristics and devices’ resistance with the probability to achieve the expected IAR (E-IAR). The mean value of E-IAR achieved seemed to be sex- and age-independent. Low- and high-resistance regimens were less likely to consent the E-IAR level (odds ratio [OR] = 0.035 and OR = 0.004, respectively). Only the basal residual volume and the inspiratory resistance, but not the Forced Expiratory Volume in 1 s (FEV1), seemed to affect the extent of IAR in asthmatic adolescents (OR = 1.131 and OR = 0.290, respectively). The results suggest that the assessment of current lung function is crucial for choosing the proper DPI for asthmatic adolescents.

## 1. Introduction

Inhalation is the most suitable and convenient route for delivering active drugs to patients suffering from airway obstruction. Respiratory drugs target the airways directly by this route, thus minimizing the drug dose and allowing a quicker onset of action [1,2]. 

The inhalation technology was greatly improved over the last years, particularly with the aim to improve its therapeutic effectiveness in more “difficult” and/or in “less compliant” patients who need long-term respiratory treatments, such as teenagers suffering from bronchial asthma. 

The effectiveness of any inhalation therapy for bronchial asthma depends on several factors [3,4,5,6]. Dry powder inhalers (DPIs) require a faster and deeper inhalation than Metered-Dose Inhalers and Soft Mist Inhalers. However, they are the most prescribed devices for daily and long-term asthma treatments [7,8]. Each DPI is characterized by its own intrinsic resistance. This resistance depends on the original engineering of each DPI and determines the pressure drop induced by patients’ inspiratory airflow rate (IAR) across the DPI itself [9]. IAR is the only active force causing the pressure drop and consequent turbulence inside the DPI that are required for disaggregating the powder to inhale. 

It was recently shown that subjects’ airflow limitations might variably affect the expected IAR across DPIs in adults and that different lung function predictors can be usefully employed, particularly in patients with asthma and other obstructive respiratory disorders [10,11]. This evidence is still missing for asthma adolescents, who usually do not comply with the inhalation procedures required for an effective asthma management. 

The present study investigated whether some parameters of lung function may predict the IAR of adolescents suffering from asthma when inhaling at different regimens of resistance.

## 2. Materials and Methods

A sample of asthma adolescents with normal cognition and dexterity referring to the CEMS Lung Unit (Verona, Italy) was consecutively recruited between June and September 2020. The lung function parameters assessed by Plethysmography Platinum DX Elite, MedGraphics, USA were: Forced Expiratory Volume in 1 s (FEV1), Inspiratory Capacity (IC), Forced Inspiratory Volume (FIV), Forced Inspiratory Flow (FIF), Total Lung Capacity (TLC), Maximal Expiratory Flow at 25% of lung filling (MEF25), Residual Volume (RV), Inspiratory Resistance (I_Raw_), and Expiratory Resistance (E_Raw_). FEV1, IC, TLC, and RV were expressed both in L and in % predicted, FIV and I_Raw_ in L/s, FIF and MEF25 in both L/s and % predicted.

The In-Check DIAL G16 (Clement Clarke International Ltd., Harlow, UK) was used for measuring IAR at three different resistance regimens, using a validated DPIs simulator capable to reproduce the patterns of intrinsic resistance that are peculiar for several DPIs. The In-Check DIAL G16 allows forced IARs ranging from 15 to 120 L/min, that correspond to the IAR levels of the majority of inhaling devices [9,10,12]. It should be taken into account that an inspiratory pressure drop of 4 kPa across the device corresponds to an inspiratory flow resistance (IFR) <0.02 kPa^0.5^min/L and requires an IAR > 100 L/min. The IFR increases to 0.020–0.040 kPa^0.5^min/L in the mid-resistance regimen and requires an IAR of 50–100 L/min. Finally, the high-resistance regimen produces an IFR > 0.040 kPa^0.5^min/L and requires an IAR < 50 L/min [3,9,13,14,15,16,17,18]. Values of IAR assessed in the present study were compared to these reference values. 

At recruitment, two expert technicians trained all subjects in the use of the In-Check DIAL simulator. Patients were randomly tested using the three different resistance regimens. Each patient performed three sequential attempts, and only the best IAR was collected for calculations (inter-measure variability ≤5%). Subjects who produced the expected IAR (E-IAR) value for each regimen [3,9,13,14,15,16,17,18] were also recorded.

Data produced by adolescents with asthma were also compared to those previously obtained from a cohort of adult asthmatics [10].

Before the investigation started, the adolescents’ parents gave their informed consent to the use of the collected information for purposes of research. 

Data are reported as means ± standard error (SE). Only the “sex” variable is reported as absolute and relative frequency. The non-parametric Wilcoxon test was used to compare adolescents and adult controls in terms of age, BMI, and lung function (the exact Fisher test was used for sex). 

For each lung function parameter, we investigated the association with the probability to achieve the E-IAR (with the three simulated resistance regimens) by using univariate logistic models. The strength of this association was measured in terms of odds ratio (OR): OR > 1 (resp. OR < 1) and suggested a positive (resp., negative) association with the tested variable, i.e., the probability to achieve the E-IAR increased (resp., decreases) as the variable’s value increased. All variables associated with the outcome (preliminary fixed at *p*-value < 0.25) were included in a multiple regression model. Finally, the best subset of predictors was selected by a backward-stepwise selection.

Finally, a sample of adult controls enrolled in our previous study [10] was used to investigate if adolescents could be associated with different inspiratory flows or with a higher probability to achieve the expected IAR. The comparison was performed using a generalized linear model (Gamma family) for inspiratory flow and a logistic regression for the probability to produce the E-IAR value.

Stata Statistical Software (Release 15) was used for all statistical calculations.

## 3. Results

Eighteen asthmatic adolescents were tested. The baseline characteristics of the sample are summarized in Table 1. The resulting lung function was compatible with bronchial asthma and (with the obvious exception of age distribution), comparable to that of a control group of adults enrolled in a previous study [10] (Table 1).

Adolescents’ IAR appeared to be inversely related to the increase of the resistance regimen (*p* < 0.001) (Table 2). The inspiratory airflows obtained in adolescents were higher than those observed in the adult controls. Such difference became progressively more evident when increasing the resistance regimen.

The results of univariate and multivariate regressions are reported in Table 3. In general, resistance regimens were highly associated with the probability to achieve the E-IAR (Table 3). In particular, when forced inspiration was performed at low- and high-resistance regimens, adolescents with asthma were less likely to reach their E-IAR compared to when they inhaled at mid-resistance: OR _Low_ vs. _mid_ = 0.035 (95% CI 0.001 to 0.84) and OR _High_ vs. _mid_ = 0.004 (95% CI < 0.001 to 0.42, respectively). These data could suggest that low-resistance DPIs are less likely to consent reaching the optimal IAR in asthmatic adolescents because the required value of IAR > 100 L/min is too high, and only some asthmatic adolescents are capable to reach this threshold (Table 3). Moreover, high-resistance DPIs seemed to be less likely to allow the optimal IARs due to the too high intrinsic resistance that has to be overcome when inhaling, while mid-resistance DPIs were the most suitable and reliable from this point of view.

In the multivariate analysis, the only lung function parameters that contributed to the prediction of IAR were RV % predicted (OR = 1.131, 95% CI 1.03 to 1.25) and I_Raw_ (OR = 0.29, 95% CI 0.09 to 0.92), whereas FEV1 was not useful (both in absolute and in % predicted values) (Table 3). 

Finally, the proportion of asthmatic adolescents who reached their E-IAR value is reported in Figure 1 for the three simulated regimens. The probability to achieve the E-IAR in asthmatic adolescents was comparable to that of adults at low- and mid-resistance regimens, while it was lower at a high-resistance regimen (OR = 0.16, 95% CI 0.03 to 0.90). 

## 4. Discussion

The delivery of respiratory drugs is substantially improved by the use of DPIs. In fact, DPIs do not contain propellants, require simplified procedures for inhalation; improve patients’ adherence to treatments, minimize the variability of the emitted dose, favor drug(s) deposition within the airways, reduce the occurrence of side effects (both local and systemic), and contribute to improve the therapeutic outcomes [4,19,20]. 

Despite these consolidated advantages, choosing the best DPI is still a critical challenge in real life [21], as the DPIs presently available perform differently in terms of inhalation and deposition patterns, according to their engineering differences [3,22,23]. Several experimental and in vitro studies extensively investigated the relative contribution of different factors, even if the available data are frequently only partially reproducible in real-life conditions [24,25,26]. 

The drivers affecting the de-aggregation and the aerosolization of dry powders to be inhaled through DPIs are the pressure drop occurring during maximal inspiratory maneuvers and the flow rate and flow acceleration generated through the device [9,27,28,29,30,31]. The interactions between volumes, flow rates, changes in pressure drops, and DPI technical peculiarities are really complex, even though it has been suggested that larger pressure drops and higher flow rates and inhaled volumes usually correspond to more effective particle dispersion and aerosolization and to a larger amount of drug(s) impacting the airways for all DPIs. [25,32,33]. DPIs can then be ranked as low-resistance (<5 Mbar^0.5^L/min^−1^, i.e., Brezhaler), mid-resistance (5–10 Mbar^0.5^L/min^−1^, i.e., Accuhaler, Diskhaler, Ellipta, Genuair; Spiromax, Clickhaler, Turbohaler, Easyhaler, Twisthaler, Nexthaler), and high-resistance inhalers (>10 Mbar^0.5^L/min^−1^, i.e., Handihaler) [12,13].

In the case of low-resistance DPIs, the inspiratory pressures needed for the effective inhalation through the DPI are suggested to work as a factor limiting the patient’s capability to generate the E-IAR level [34], even if in some studies [3,35,36], but not in others [10,30,33], patient’s age and gender were described as the unique variables able to influence IAR through DPIs.

The patient’s respiratory disorder and the corresponding lung function limitations were considered trivial factors in this regard, though airway and parenchymal structures can be variably and peculiarly compromised, and their mechanical performance limited [37]. It should be emphasized that the patient’s IAR is the only force generated during inhalation. This drive is required (1) to produce the necessary pressure drop, (2) to elicit turbulence inside the DPI, and (3) to produce the disaggregation, the micro-dispersion and, finally, the delivery of the powdered drug [1,9,32]. However, even if specific data are very few, it has been shown that the IAR achievable through a DPI is related to the square root of the occurring pressure drop and that the dose of the inhaled drug targeting the airways is directly related to the IAR increase [6,32,38].

The role of a subject’s lung function, though minimized in some studies (particularly, in bronchial asthma) should be much more valued in our opinion, as different degrees of flow limitation can variably correspond to effects on patients’ inspiratory/expiratory performances. Unfortunately, the relationships linking pressure drops, IAR, and DPI resistance have not been frequently studied in asthma patients, likely because a complete plethysmography assessment of lung function parameters does not represent a usual procedure in clinical practice, particularly for asthmatic adolescents. Nevertheless, each lung function parameter provides, even if to a variable extent, a physiological sign, and the occurrence of peculiar respiratory limitations could be able to affect the effective use of DPIs. We emphasize that a DPI choice based only on FEV1 values, even if simple to obtain, failed to predict the effective IAR through DPIs, as FEV1 is characterized by a too low specificity [39,40,41]. It is presumable that other more appropriate parameters should be carefully assessed in order to unveil more specific lung function limitations affecting IAR when inhaling throughout DPIs at different regimens of intrinsic resistance.

The data of the present investigation suggest that the extent of IAR through DPIs in asthmatic adolescents could be affected by the subjects’ pattern of airflow limitation. In other words, changes in airflow limitation may variably affect the inspiratory airflow rate required to overcome the proper resistance of different DPIs, and, consequently, to assure the effective delivery of the powdered drug(s) within the airways. Our results also showed that only mid-resistance DPIs consented to achieve the most convenient IAR with the highest frequency in asthmatic adolescents. In other words, the majority of asthmatic adolescents can achieve their E-IAR only by inhaling through a mid-resistance DPI, regardless of age, sex, and BMI. As already observed in adult asthmatics [10], similar performances cannot be obtained when inhaling through low- and high- resistance DPIs (Figure 1).

In addition to DPI resistance, RV and I_Raw_ were the only predictors of E-IAR in asthmatic adolescents, that is, in those individuals characterized by a not negligible reduction in their airway patency. The variable proportion of young patients who achieved their E-IAR seemed to mirror the effect of their respiratory condition.

Some limitation likely biased the present investigation. Data were obtained from a monocentric sample of patients, with a limited size. However, the results are consistent with those of a previous analysis focused on the adult population [10], and an extension of the present analysis, aimed to increase the sample size, is already in progress. The measurements of DPIs’ resistance were carried out by the In-Check DIAL G16, used as a simulator, and the results were assumed to be related to DPIs’ intrinsic resistance.

The points of strength of this study are: (1) the adolescents’ performance in terms of E-IAR was investigated in real life for the first time, (2) possible predictors of the proper IAR were explored for the first time in asthmatic adolescents using a quite exhaustive set of lung function parameters, (3) proper statistical models were used.

## 5. Conclusions

The assessment of DPIs is still a challenge in respiratory medicine, particularly in asthmatic adolescents, i.e., those individuals who are the least compliant with any regular and long-term inhalation treatment. In general, the engineering peculiarities of DPI and their intrinsic resistive regimens can variably affect the extent of patients’ IAR. Lung function further contributes to affect IAR per se, even in asthma adolescents. Only some lung function parameters, such as RV and I_Raw_ but not FEV1, seem to predict the E-IAR through DPIs in asthmatic adolescents.

The analytic assessment of lung function is strongly suggested for asthmatic adolescents, with the aim to more effectively personalize the choice of DPI.

## Figures and Tables

**Figure 1 children-09-00377-f001:**
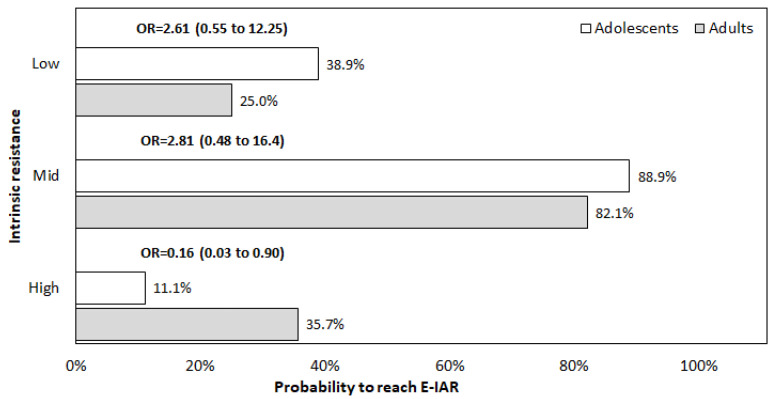
Proportion of subjects who reached their E-IAR value at the three simulated regimens. E-IAR: expected inspiratory airflow rate; OR: odds ratio adjusted for RV and I_Raw_.

**Table 1 children-09-00377-t001:** Mean ± standard error of lung function parameters measured in the sample of adolescents and in the adult controls (absolute and relative frequency were used for the variable “sex”).

Variables	Adolescents	Retrospective Adult Controls	*p*-Value
N	18	28	
Sex (% female)	10 (55.6%)	17 (60.7%)	0.4820
Age	16.9 ± 0.39	52.1 ± 2.89	<0.0001
BMI	22.1 ± 0.44	25.9 ± 1.15	0.0229
FEV1 (L)	3.1 ± 0.07	2.8 ± 0.16	0.1315
FEV1 (% pred)	97.6 ± 1.32	93.5 ± 2.91	0.3216
IC (L)	3.1 ± 0.10	2.9 ± 0.14	0.3368
IC (% pred)	104.3 ± 1.76	107.4 ± 4.12	0.9312
FIV (L)	3.4 ± 0.21	3.3 ± 0.19	0.8728
FIF max (L/s)	4.9 ± 0.45	4.9 ± 0.36	>0.9999
FIF max (% pred)	84.1 ± 4.48	79.9 ± 4.82	0.6845
MEF25 (L/s)	1.9 ± 0.12	1.4 ± 0.14	0.0164
MEF25% (% pred)	95.2 ± 3.59	81.4 ± 6.22	0.0165
TLC (L)	5.8 ± 0.23	5.6 ± 0.17	0.3976
TLC (% pred)	98.6 ± 2.36	95.7 ± 2.93	0.3358
RV (L)	1.5 ± 0.08	1.8 ± 0.14	0.0425
RV (% pred)	88.8 ± 2.82	91.1 ± 6.59	0.4867
I_Raw_ (L)	1.8 ± 0.18	3.1 ± 0.57	0.0717

FEV1: Forced Expiratory Volume in 1 s; IC: Inspiratory Capacity; FIV: Forced Inspiratory Volume; FIF: Forced Inspiratory Flow; TLC: Total Lung Capacity; MEF25: Maximal Expiratory Flow at 25% of lung filling; RV: Residual Volume; I_Raw_; Inspiratory Resistance.

**Table 2 children-09-00377-t002:** Mean inspiratory flow (L/min) ± standard error measured at the three resistance regimens.

DPI Resistance	Adolescents	Retrospective Adult Controls	Mean Difference (95% CI)
Low	90.56 ± 3.75	86.96 ± 3.87	3.59 (−6.84 to 14.02)
Mid	76.67 ± 3.38	70.89 ± 3.66	5.77 (−3.78 to 15.33)
High	64.44 ± 2.77	53.21 ± 3.37	11.23 (2.81 to 19.65)
Non parametric test for trend	*p* < 0.001	*p* < 0.001	

**Table 3 children-09-00377-t003:** Results of univariate and multivariate logistic regression (after stepwise selection).

Variables	Univariate Model OR (95% CI)	Multivariate Model OR (95% CI)
Sex (male)	1.204 (0.67 to 2.16)	
Age (years)	1 (0.98 to 1.02)	
BMI	1.01 (0.96 to 1.06)	
FEV1 (L)	0.969 (0.71 to 1.32)	
FEV1 (%)	1.002 (0.98 to 1.02)	
IC (L)	1.006 (0.69 to 1.46)	
IC (%)	1.003 (0.99 to 1.02)	
FIV (L)	1.024 (0.71 to 1.47)	
FIF (L/s)	0.98 (0.83 to 1.15)	
FIF (%)	1.002 (0.99 to 1.02)	
MEF25 (L/s)	1.026 (0.75 to 1.41)	
MEF25 (%)	0.996 (0.99 to 1)	
TLC (L)	1.147 (0.86 to 1.53)	
TLC (%)	1.011 (0.99 to 1.03)	
RV (L)	1.222 (0.85 to 1.76)	
RV (%)	1.006 (1 to 1.01)	1.131 (1.03 to 1.25)
I_Raw_ (L)	1.027 (0.97 to 1.09)	0.290 (0.09 to 0.92)
DPI Resistance		
Low vs. mid	0.086 (0.02 to 0.36)	0.035 (0.001 to 0.84)
High vs. mid	0.116 (0.03 to 0.41)	0.004 (<0.001 to 0.42)

FEV1: Forced Expiratory Volume in 1 s; IC: Inspiratory Capacity; FIV: Forced Inspiratory Volume; FIF: Forced Inspiratory Flow; TLC: Total Lung Capacity; MEF25: Maximal Expiratory Flow at 25% of lung filling; RV: Residual Volume; I_Raw_; Inspiratory Resistance; OR: odds ratio; CI: confidence interval.

## Data Availability

Authors do not wish to share their data without their permission.

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
