# Peer review of "Lung Function Can Predict the Expected Inspiratory Airflow Rate through Dry Powder Inhalers in Asthmatic Adolescents"

_children, 2022, doi:10.3390/children9030377_

Round 1

Reviewer 1 Report

Line 35 " defectinveness" is the word effectiveness 

Expand the work DPI when first used in the manuscript.

Line 44 to 45- sentences needs to be revised to bring clarity. 

LIne 50 to 53-" peculiar changes " can be delected.  " asthma adolescents" needs to be adolescents  suffering from asthma. 

Line 53 and 54- what is normal ability? and what is manual skills?

Line 109. How are the lungs functions compatible with bronchial asthma ?

What is the significance of logistic regression model of FEV1 and FEV1% ( both are same meaning) 

Author Response

  • Line 35 " defectinveness" is the word effectiveness:
    • the typo was corrected
  • Expand the work DPI when first used in the manuscript:
    • done
  • Line 44 to 45- sentences needs to be revised to bring clarity:
    • the sentence has been reworded. 
  • LIne 50 to 53-" peculiar changes " can be delected.  " asthma adolescents" needs to be adolescents  suffering from asthma:
    • done 
  • Line 53 and 54- what is normal ability? and what is manual skills?
    • the sentence has been partially modified to clarify the meaning
  • Line 79: we caught a typing error in line 79:
    • “subjects” has been corrected.
  • Line 109. How are the lungs functions compatible with bronchial asthma?
    • The distribution of almost all respiratory parameters was coherent with the pathology under study. However, we rephrase the sentence and the meaning was clarified
  • What is the significance of logistic regression model of FEV1 and FEV1% (both are same meaning): All covariates were tested in univariate regression (i.e. one by one) in order to highlight which variables were the most associated with the expected inspiratory airflow rate (see Table 3). We used both absolute and % of predicted value for FEV1, IC, FIF, TLC and RV as these parameters not always show the same association with outcomes under study.

Reviewer 2 Report

This manuscript aimed to investigate whether lung function limitations occurring in asthma adolescents affect their IAR when inhaling through a DPI simulator reproducing different resistance regimens. The topic is relevant and the study design and the study procedure are very clear. The article has a clear language and the aim of the study it is clear and interesting, however the study has a very small sample, so the authors must be careful when interpreting the results. I recommend that the authors increase the sample. 

I have, also, suggestions for revision:

In the abstract, the authors have used some abbreviations that need to be spelled out, such as DPIs and  FEV-1.

Captions on figures should appear below the figure and not above it. Authors should change the caption of figure 1 to the under of it.

Author Response

This manuscript aimed to investigate whether lung function limitations occurring in asthma adolescents affect their IAR when inhaling through a DPI simulator reproducing different resistance regimens. The topic is relevant and the study design and the study procedure are very clear. The article has a clear language and the aim of the study it is clear and interesting, however the study has a very small sample, so the authors must be careful when interpreting the results. I recommend that the authors increase the sample. 

ANSWER: First of all, we thank Reviewer 1 for defining our paper as following:  “The topic is relevant and the study design and the study procedure are very clear. The article has a clear language and the aim of the study it is clear and interesting”. We can ensure the Reviewer 1 that the extension of asthma adolescents  is already in progress.

I have, also, suggestions for revision:

  • In the abstract, the authors have used some abbreviations that need to be spelled out, such as DPIs and  FEV-1: done
  • Captions on figures should appear below the figure and not above it. Authors should change the caption of figure 1 to the under of it: done

Round 2

Reviewer 1 Report

We appreciate the authors inclusion of the suggested changes. The size of the sample still remains the main limitation to draw meaningful conclusion. 

Author Response

Dear Reviewer, 

We add more details in the discussion about the small sample size. Moreover we weakened some sentences in the Results and Discussion sections due to the small sample size analyzed.

Reviewer 2 Report

My biggest concern is the small sample; however, the authors ensure that the extension of asthma adolescents is already in progress, whereby  I only suggest to add this information in the discussion.

Author Response

Dear Reviewer, 

thank you for the suggestion. We added the information about the ongoing study iun the discussion. Moreover we weakened some sentences in the Results and Discussion sections due to the small sample size analyzed.

This manuscript is a resubmission of an earlier submission. The following is a list of the peer review reports and author responses from that submission.